# Investigating the Temperature Shock of a Plate in the Framework of a Static Two-Dimensional Formulation of the Thermoelasticity Problem

Andry Sedelnikov [1,*] , Valeria Serdakova [2], Denis Orlov [1,*] and Alexandra Nikolaeva [3]

1   Department of Space Engineering, Samara National Research University, 443086 Samara, Russia
2   Department of Higher Mathematics, Samara National Research University, 443086 Samara, Russia; valeriay121@yandex.ru
3   Department of Theoretical Mechanics, Samara National Research University, 443086 Samara, Russia; ezhevichka333@gmail.com
*   Correspondence: axe_backdraft@inbox.ru (A.S.); grand_99v@mail.ru (D.O.)

**Abstract:** The paper investigates the stress–strain state of a homogeneous rectangular plate after a temperature shock. It is believed that the plate is the first approximation of the solar panel model of a small spacecraft. To study the stress–strain state of the plate, a two-dimensional thermoelasticity problem is posed. The problem has a static formulation, since it does not take into account the dynamics of the plate's natural oscillations. These oscillations affect the stress–strain state through the initial deflection of the plate at the time of the temperature shock. This deflection changes the parameters of the temperature shock and does not allow the use of a one-dimensional formulation of the thermoelasticity problem. As a result of solving the static two-dimensional thermoelasticity problem, approximate solutions are obtained for the components of the plate point's displacement vector after the temperature shock. An approximation of the temperature field is presented. A numerical simulation is carried out. The correspondence of the obtained approximate analytical dependencies of the components of the plate point's displacement vector to the numerical simulation data is analyzed. The proposed method can be used to assess the significance of the influence of the small spacecraft's solar panels temperature shock on the dynamics of its rotational motion.

**Keywords:** temperature shock; stress–strain state; two-dimensional thermoelasticity problem; solar panel; small spacecraft

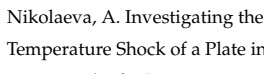



## 1. Introduction

The temperature shock problem plays an important role in modern space technology. This is especially true for small spacecrafts, which are becoming increasingly common for solving various tasks in space [1,2]. There are several areas that are important for the development of the space industry in which it is necessary to study and take into account the temperature shock for the qualitative fulfillment of the target tasks.

The first area is remote sensing of the Earth. The orientation accuracy of the Earth's remote sensing spacecraft may be impaired due to the temperature shock [3,4]. Thermal structural disturbances are investigated in [5]. They arise as a result of uneven solar heating. According to the authors [5], these disturbances can not only reduce the accuracy of the spacecraft guidance, but also affect the stability of its motion. The deterioration of functional characteristics as a result of the temperature shock for the Hubble Space Telescope and the UARS upper atmosphere research satellite is noted in [6]. The results of analytical and experimental studies of the temperature shock are presented in [7,8]. They show the importance of temperature differences in thickness and their time derivatives for temperature deformations. The relations between characteristic thermal and structural response times in thermally induced structural movements of the spacecraft solar panels are given.

Thermal structural tests have demonstrated significant quasi-static bending deformations of solar panels under temperature shock in laboratory conditions. Stricter requirements for the functional characteristics of the Earth's remote sensing spacecraft imply an even more accurate solution to orientation problems. Additionally, this is practically unattainable without taking into account and studying the temperature shock.

The second area is space technology. At the moment, there are no implemented projects for small technological spacecrafts. Perturbations arising from the temperature shock create unacceptably high micro-accelerations [9,10]. These micro-accelerations do not allow the successful implementation of gravity-sensitive processes on board a small spacecraft [11]. In [9], the design and layout scheme of a small spacecraft with a single solar panel are investigated. It is shown that micro-accelerations from the temperature shock can exceed the permissible values by an order of magnitude [9]. In [3], the temperature shock problem for a symmetrical small spacecraft is considered. In this case, part of the perturbations will be compensated. However, the remaining ones still affect the micro-acceleration field of the internal environment of the small spacecraft. It is noted in [3] that symmetry breaking will cause greater perturbations than for the symmetrical small spacecraft. It is shown in [12] that it is possible to deal with perturbations from the temperature shock with the help of the executive bodies of the motion control system. Therefore, the study of the temperature shock and the development of effective algorithms for neutralizing its impact on small spacecrafts can ensure progress in the field of space materials science.

The third area is the technology for cleaning space debris. Various methods of de-orbiting spacecrafts after the expiration of their active existence have been developed [13,14]. One of such methods is the use of cable systems [15]. When using cable systems to transport inactive small spacecraft with solar panels, a temperature shock may occur. In an unfavorable combination of circumstances, disturbances from the temperature shock can contribute to the separation of the cable from space debris. Thus, the temperature shock can affect the effectiveness of some cleaning up space debris methods in the form of inactive small spacecrafts with solar panels.

We should also mention the experience of deploying new-generation ROSA solar panels on board the International Space Station [16]. Ultra-thin ROSA panels save up to 20% of their mass with comparable energy outputs [12]. However, the thermal oscillations resulting from the temperature shock were very significant. When installing such panels on the small spacecraft, the issue of their controllability will become acute. During the experiment on the International Space Station, the ROSA panel could not be rolled back due to intense thermal oscillations [16]. It was undocked from the International Space Station in an expanded form. This once again confirms the relevance of studying temperature shocks for the successful implementation of space missions.

The assessment of the temperature shock effect on the motion of the small spacecraft within the framework of the one-dimensional model of thermal conductivity is a point estimation [4,17]. It considers the most dangerous case from the point of view of the intensity of the temperature shock impact. At the same time, the solar panel had a flat shape at the time of the temperature shock [4,17]. The normal to this plane is parallel to the incident solar flux. This assessment allows us to answer the question of whether it is necessary to take into account the temperature shock effect on the motion of a small spacecraft. There is no need to take into account the temperature shock for medium-class spacecraft and the class of orbital space stations. Its influence on the motion of these classes of spacecrafts is negligible compared to other disturbing factors [9,12].

With a more detailed study of the temperature shock, the one-dimensional statement may not be sufficient. The solar panels of the small spacecraft make their own oscillations due to the perturbation effect. Taking into account the dynamics of oscillations will lead to the need to solve the temperature shock problem in a general three-dimensional formulation. However, in a number of cases, the solution of a two-dimensional problem may be sufficient. It consists of taking into account the initial deflection of the panel at the time of the temperature shock onset (Figure 1) [17].

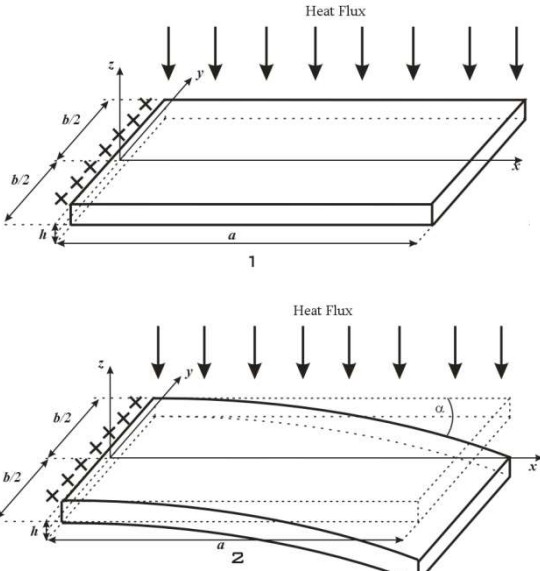

**Figure 1.** The panel at the moment of temperature shock: 1 without initial deflection; 2 with initial deflection [17].

In [18], it is shown that this approach significantly complicates the problem being solved. In [18], as well as in [19,20], the cases of stability loss of the solar panel due to the combined effect of initial deflection stresses and thermal stresses were investigated.

In [21,22], the design of aerospace engineering products is considered, taking into account their thermoelastic properties.

In this paper, we have attempted to obtain approximate analytical dependencies of the components of the plate point's displacement vector within the framework of the two-dimensional model of thermoelasticity. A similar problem for the one-dimensional model was solved in [4]. Therefore, a comparative analysis of the solutions obtained for one-dimensional and two-dimensional models of thermoelasticity will be carried out. This approach makes it possible to use a two-dimensional formulation without significantly complicating the mathematical model of the temperature shock. At the same time, the convergence of the results with the data of the computational experiment is ensured.

The article is organized as follows: at first, simplified assumptions are introduced for constructing a two-dimensional model of thermoelasticity, then a mathematical formulation of the third initial boundary value problem is given. As a result of its solution, approximate dependencies of the components of the plate point's displacement vector are obtained. Then, the obtained results are compared with the results of a computational experiment, and the final conclusions are made.

## 2. Theoretical Part: Simplifying Assumptions in the Construction of the Two-Dimensional Model of Thermoelasticity

We will abandon some of the simplifying assumptions of the one-dimensional formulation presented in [4]. We will leave some of the assumptions unchanged. In this case, it will be possible to correctly conduct a comparative analysis of the results.

1.  The model of a body exposed to the temperature shock is a thin, homogeneous plate.
2.  The boundary conditions for the thermoelasticity problem are rigid sealing on one edge and three free other edges of the plate.

The preservation of these hypotheses will allow us to maintain the structure of the one-dimensional thermoelasticity problem solution [4]. In particular, representations of the components of the plate point's displacement vector are valid: $u_x = 0$; $u_y = u_y(x, y, t)$ and $u_z = u_z(x, t)$.

3.  At the moment of the temperature shock onset ($t = 0$), the plate has an arbitrary shape that does not violate the deflection structure:

$$u_{z0} = u_{z0}(x, 0).$$

The boundary conditions and the initial shape of the plate can determine the deflection structure in different ways. However, a more general structure, $u_{z0} = u_{z0}(x, y, 0)$, will lead to the need for a three-dimensional formulation of the problem. Therefore, the considered two-dimensional approach will be incorrect. In a real situation, many characteristic sizes of the solar panels of various small spacecrafts allow us to neglect transverse oscillations compared with longitudinal ones [18,23]. Therefore, this hypothesis has a basis.

4.  We consider the solar radiation flux to be uniform and stationary, with the maximum possible power at the level of the Earth's orbit relative to the Sun.

In the two-dimensional formulation, this value determines the maximum value of the heat flux. In the one-dimensional formulation [4], this value is the same for each section of the plate. However, now, due to the initial deflection of the plate (Figure 1), the heat flux will not have a constant value at each point of its surface layer.

5.  The plate's natural oscillations statically affect their own temperature field.

The static effect of oscillations in this work means the effect on the temperature field by means of the initial deflection of the plate at the time of the temperature shock onset (Figure 1). The dynamic effect of oscillations is associated with a change in the position of the plate as a result of oscillations during the temperature shock. It is excluded from this statement of the problem. It is believed that the oscillation process is significantly slower than the temperature shock process.

6.  The plate point's displacement vector has the following form: $\vec{u} = \left(0, u_y, u_z\right)$.

It is assumed that the points movements in the direction of the longitudinal *x*-axis are small compared to movements along other axes. This is especially true when compared with deflections $u_z$.

7.  All thermophysical properties of the plate are assumed to be homogeneous and the same throughout the temperature range.
8.  When describing thermal conductivity, Fourier's law is valid.
9.  The initial temperature distribution field of the plate is considered homogeneous.
10. The thickness of the plate is negligible compared to its length and width.
11. The heat exchange through the side surfaces of the plate is negligible compared to the heat exchange through the upper and lower surfaces.

The last five hypotheses have not changed in comparison with the one-dimensional formulation [4].

12. The initial deflection should be such as to prevent the stability loss of the plate in the process of temperature shock.

The two-dimensional formulation significantly expands the range of simulated situations in comparison with the one-dimensional formulation. For example, it becomes possible to simulate the temperature shock at an arbitrary value of the angle between the direction of the heat flux and the normal to the plate section. The significant value of this angle may not only be associated with the initial deflection of the plate. It can be caused in a real situation, for example, by the position of the solar panel plane relative to the Sun at the time when the small spacecraft exits from the shadow of the Earth. This situation is associated with the rotational motion of the small spacecraft relative to the center of mass.

The statement of the third initial boundary value problem of two-dimensional thermoelasticity.

Let the plate at the moment of temperature shock have an arbitrary curved shape corresponding to the simplifying assumption 3 (Figure 2).

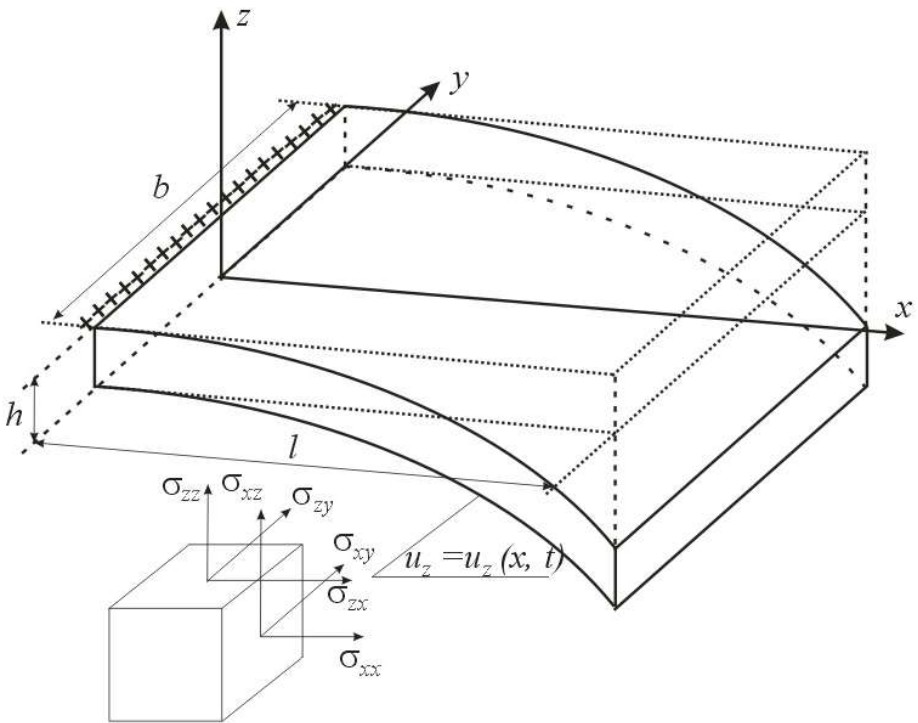

**Figure 2.** The shape of the plate at the moment of temperature shock.

In this case, the third initial boundary value problem of two-dimensional thermal conductivity will have the following form [18]:

$$
\begin{cases}
\frac{\partial T(x, z, t)}{\partial t} = a^2 \left( \frac{\partial^2 T(x, z, t)}{\partial x^2} + \frac{\partial^2 T(x, z, t)}{\partial z^2} \right), \ 0 \le x \le l, \ h \le z \le 0, \ t > 0; \\
\left( \lambda \frac{\partial T(x, h, t)}{\partial n} \right) = Q_0 \cdot \cos \left( \frac{du_{z0}(x, 0)}{dx} \right) - e\Theta\left( T^4(x, \ h, \ t) - T_C^4 \right), \ 0 \le x \le l, z = h, \ t > 0; \\
\left( \lambda \frac{\partial T(x, 0, t)}{\partial n} \right) = -e\Theta\left( T^4(x, \ 0, \ t) - T_C^4 \right), \ 0 \le x \le l, z = 0, \ t > 0; \\
T(x, \ z, \ 0) = T_0 = \ const.
\end{cases}
\tag{1}
$$

where $T = T(x, z, t)$ is the temperature field of the plate; $a$ is the temperature conductivity coefficient; $\lambda$ is the thermal conductivity coefficient; $\vec{n}$ is the unit vector of the external normal to the plate surface element; $Q_0$ is the maximum value of the incident heat flux; $e$ is the degree of blackness of the plate material; $\Theta$ is the Stefan–Boltzmann constant; and $T_C$ is the temperature of the environment surrounding the plate.

For clarity of comparing the results with the one-dimensional model [4] in (1), it is assumed that the incident heat flux is perpendicular to the surface of the plate near the seal ($x = 0$).

Let us further consider the deflections of the plate. When setting the thermoelasticity problem, we use the Sophie Germain equation for thin plates [24]:

$$
D \frac{\partial^4 u_z(x, \ t)}{\partial x^4} + \rho \, h \frac{\partial^2 u_z(x, \ t)}{\partial t^2} = -2\mu\alpha \int_{-h/2}^{h/2} \left[ 2 \frac{\partial T(x,z, \ t)}{\partial z} + z \frac{\partial^2 T(x,z, \ t)}{\partial z^2} \right] dz + \frac{\partial^2 u_{z0}(x, \ 0)}{\partial x^2} \sigma_{xz},
\tag{2}
$$

$$0 \le x \le l, \ t > 0$$

where $\sigma_{xz}$ is the stress tensor component shown in Figure 2.

It can be noted that in comparison with the one-dimensional model (Equation (5) [4]), there is an initial deflection $u_{z0} = u_{z0}(x, 0)$ and the stresses caused by it $\sigma_{xz}$ in the right part (2). The boundary conditions will have the form (simplifying assumption 2).

The fixed edge $x = 0$ (hard sealing conditions) geometric boundary conditions:

$$\begin{cases} u_z(0, \ t) = 0, \ \ x = 0, \ \ t > 0; \\ \frac{\partial u_z(x, \ t)}{\partial x} = 0, \ \ x = 0, \ \ t > 0. \end{cases} \tag{3}$$

The free edge $x = l$ static boundary conditions:

$$\begin{cases} \frac{\partial^2 u_z(x, \ t)}{\partial x^2} + v \frac{\partial^2 u_z(x, \ t)}{\partial y^2} = 0, \ \ x = l, \ \ t > 0; \\ \frac{\partial^3 u_z(x, \ t)}{\partial x^3} + (2 - v) \frac{\partial^3 u_z(x, \ t)}{\partial x \, \partial y^2} = 0, \ \ x = l, \ \ t > 0, \end{cases} \tag{4}$$

Because $u_z = u_z(x, t)$ (4) can be converted as follows:

$$\begin{cases} \frac{\partial^2 u_z(x, \ t)}{\partial x^2} = 0, \ \ x = l, \ \ t > 0; \\ \frac{\partial^3 u_z(x, \ t)}{\partial x^3} = 0, \ \ x = l, \ \ t > 0. \end{cases} \tag{5}$$

Let us supplement Equation (2) with the initial conditions:

$$u_z(x, \ 0) = u_{z0}(x, \ 0), \ \ 0 \leq x \leq l, \ \ t = 0. \tag{6}$$

In this case, the function $u_{z0} = u_{z0}(x, 0)$ is considered to be set or it can be determined from the analysis of the process of the plate's natural oscillations.

Let us consider another component of the displacement vector $u_y(x, y, t)$. Its structure is more complex than the deflections structure $u_z = u_z(x, t)$. This is typical even for the one-dimensional statement [4]. Let us assume that the plate has come to equilibrium after the temperature shock and write down the equilibrium equation [4,24]:

$$\frac{3(1 - v)}{1 + v} grad \ div \ \vec{u} \ - \ \frac{3(1 - 2v)}{2(1 + v)} \ rot \ rot \ \vec{u} = \alpha \ grad \ T, \ \ t = \infty. \tag{7}$$

In Equation (7), the moment of equilibrium is conditionally marked as $t = \infty$. Let us write Equation (7) in scalar form:

$$\begin{cases} \frac{\partial^2 u_y(x, \ y, \ t)}{\partial x \, \partial y} = \alpha \, \frac{\partial T(x, \ z, \ t)}{\partial x}, \ \ 0 \leq x \leq l, \ 0 \leq y \leq \frac{b}{2}, \ \ 0 \leq z \leq h, \ \ t = \infty; \\ \frac{3(1-v)}{1+v} \frac{\partial^2 u_y(x, \ y, \ t)}{\partial y^2} + \frac{3(1-2v)}{2(1+v)} \frac{\partial^2 u_y(x, \ y, \ t)}{\partial x^2} = 0, \ \ 0 \leq x \leq l, \ 0 \leq y \leq \frac{b}{2}, \ \ t = \infty; \\ \frac{3(1-2v)}{2(1+v)} \frac{\partial^2 u_z(x, \ t)}{\partial x^2} = \alpha \, \frac{\partial T(x, \ z, \ t)}{\partial z}, \ \ 0 \leq x \leq l, \ 0 \leq y \leq \frac{b}{2}, \ \ 0 \leq z \leq h, \ \ t = \infty. \end{cases} \tag{8}$$

In contrast to [4], the first equation of the system (8) has a nonzero right side. This is because the temperature field depends on the longitudinal coordinate $x$. This situation changes the approach to the solution. The decomposition proposed in [4]:

$$v(x, \ y, \ t) = v_1(x, \ t) + y \, v_3(t) \tag{9}$$

does not satisfy the first equation of the system (8). However, this equation imposes a restriction on the function $u_y = u_y(x, y, t)$. It should depend linearly on $y$. Then, after taking the partial derivative with respect to $y$, this coordinate disappears from the dependence. However, even after that, the left and right parts of the first equation of the system (8) will not be consistent in the composition of the variables. Indeed, the left part does not depend on the coordinate $z$. To remove such a contradiction, let us soften the requirements for the first Equation (8). Let us refer to the simplifying assumption 1 and request that the first

Equation (8) is satisfied only for the median surface of the plate $z = h/2$. Then, the system (8) will have the following form:

$$
\begin{cases}
\frac{\partial^2 u_y(x,\, y,\, t)}{\partial x\, \partial y} = \alpha\, \frac{\partial T(x,\, z,\, t)}{\partial x}, & 0 \le x \le l,\ 0 \le y \le \frac{b}{2},\ z = \frac{h}{2},\ t = \infty; \\[2mm]
\frac{3(1-2\nu)}{2(1+\nu)} \frac{\partial^2 u_y(x,\, y,\, t)}{\partial x^2} = 0, & 0 \le x \le l,\ 0 \le y \le \frac{b}{2},\quad t = \infty; \\[2mm]
\frac{3(1-2\nu)}{2(1+\nu)} \frac{\partial^2 u_z(x,\, t)}{\partial x^2} = \alpha\, \frac{\partial T(x,\, z,\, t)}{\partial z}, & 0 \le x \le l,\ 0 \le y \le \frac{b}{2},\ 0 \le z \le h,\ t = \infty.
\end{cases}
\tag{10}
$$

The third equation of the system (10) requires a linear dependence of the temperature field on $z$. In this case, system (10) is correct and consistent in the composition of variables.

Let us write down the Dalembert principle for the forces acting on the points of the plate in the projection on the $y$ axis [4]:

$$
\frac{3(1-2\nu)}{2(1+\nu)} \frac{\partial^3 u_y(x,y,t)}{\partial x^3} + \frac{6\rho\left(1-\nu^2\right)}{Eh^2} \frac{\partial^2 u_y(x,y,\, t)}{\partial t^2} b = \frac{12\left(1-\nu^2\right)}{h^3} \alpha \int_0^h [T(x,\ z,\ t) - T_0]\, dz,
$$
$$
0 \le x \le l,\ 0 \le y \le \frac{b}{2},\ t > 0
\tag{11}
$$

Equation (11) describes the deformation process, the final stage of which is the second equation of the system (10). Let us add boundary and initial conditions. The fixed edge $x = 0$ (hard sealing conditions) geometric boundary conditions:

$$
\begin{cases}
u_y(0, y,\ t) = 0, & x = 0,\ 0 \le y \le \frac{b}{2},\ t > 0; \\[2mm]
\frac{\partial u_y(x,\, y,\, t)}{\partial x} = 0, & x = 0,\ 0 \le y \le \frac{b}{2},\ t > 0.
\end{cases}
\tag{12}
$$

The free edge $x = l$ static boundary conditions:

$$
\begin{cases}
\frac{\partial^2 u_y(x,y,\, t)}{\partial x^2} + \nu \frac{\partial^2 u_y(x,y,\, t)}{\partial y^2} = 0, & x = l,\ t > 0; \\[2mm]
\frac{\partial^3 u_y(x,y,\, t)}{\partial x^3} + (2 - \nu) \frac{\partial^3 u_y(x,y,\, t)}{\partial x\, \partial y^2} = 0, & x = l,\ t > 0.
\end{cases}
\tag{13}
$$

Let us take into account the requirements of linear dependence $u_y = u_y(x, y, t)$ from $y$ and transform (13) to the form:

$$
\begin{cases}
\frac{\partial^2 u_y(x,y,\, t)}{\partial x^2} = 0, & x = l,\ t > 0; \\[2mm]
\frac{\partial^3 u_y(x,y,\, t)}{\partial x^3} = 0, & x = l,\ t > 0.
\end{cases}
\tag{14}
$$

As an initial condition, let us choose the absence of initial deformations of the plate in the direction of the $y$ axis [4]:

$$
u_y(x,\ y,\ 0) = 0,\ 0 \le x \le l,\ 0 \le y \le \frac{b}{2},\ t = 0.
\tag{15}
$$

Thus, the vector of the plate's initial deformations in the considered formulation is limited and will have the following form: $\vec{u}_0 = (0,\ 0,\ u_{z0})$.

The derivation of approximate analytical dependencies for the components of the displacement vector.

First of all, let us consider deflection as the most significant deformation of the plate points. The analysis of Equation (2) shows that the one-dimensional solution can be used for deflection [4]. However, the initial deflection should be taken into account:

$$
u_z(x,\ t) = \frac{At}{t + a}\left(x^4 - 4lx^3 + 6l^2 x^2\right) + u_{z0}(x, 0),\ 0 \le x \le l,\ t > 0.
\tag{16}
$$

In the expression (16), the first term of the right part was obtained in [4] within the framework of solving the one-dimensional thermoelasticity problem. The second term

$u_{z0} = u_{z0}(x, 0)$ must satisfy the boundary conditions (3) and (5) and represent the plate's own shape during its oscillations [25].

The complication of the type of the temperature field dependence on the coordinates leads to the need to revise the approximate dependence for the temperature field in comparison with the one-dimensional thermoelasticity problem. Figure 3 shows the temperature field for the one-dimensional [4] and two-dimensional problems. The simulation was carried out in the ANSYS package.

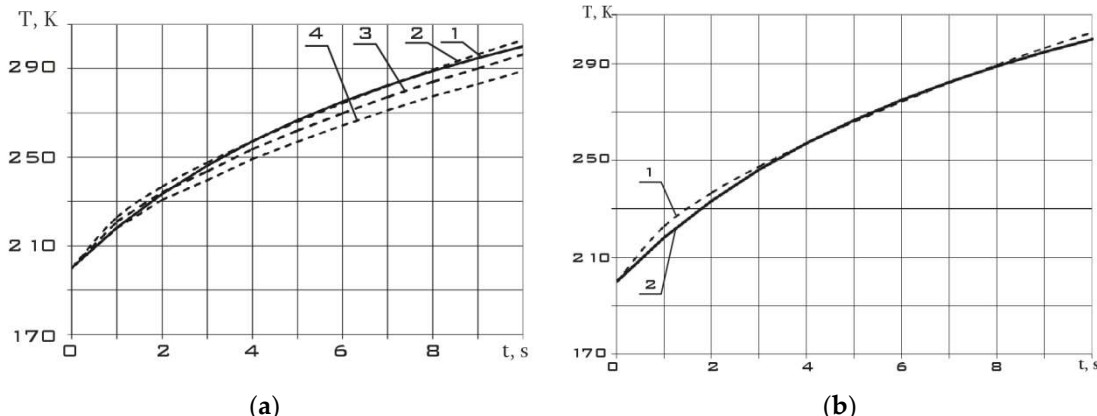

| (a) | (b) |

**Figure 3.** Dynamics of the temperature field: (**a**) For the one-dimensional thermoelasticity problem (cited [4], Figure 2). 1 a computational experiment; 2 an approximate dependence. (**b**) For the two-dimensional thermoelasticity problem: 1 an approximate dependence; 2 near the seal (x ≈ 0); 3 in the middle of the plate (x ≈ 1/2); 4 on the free edge of the plate (x ≈ l).

Figure 3 shows that the approximate dependence proposed in [4] for the temperature field ($C$ = 200 K/m, $a$ = 1 s):

$$T(z, \ t) = Cz\frac{t}{t + a} + T_0, \ \ 0 \leq z \leq h, \ \ t > 0 \tag{17}$$

is consistent with the data of the computational experiment only near the seal. This is explained by the fact that in the two-dimensional formulation that we are considering, the incident flux near the seal approximately coincides with the normal to the surface of the plate. In this paper, the following temperature dependence is proposed:

$$T(x, \ z, \ t) = Cz\frac{t}{t + a} + f(x, \ t) + T_0, \ \ 0 \leq x \leq l, \ \ 0 \leq z \leq h, \ \ t > 0. \tag{18}$$

In this case, the partial derivative with respect to $z$ of expressions (17) and (18) will be the same. This ensures the identical equality of integrals on the right side of Equation (2) and the similar equation for the one-dimensional problem (Equation (5) [4]). Therefore, it is correct to say that the difference between approximate solutions for the two-dimensional problem (16) and the corresponding one-dimensional problem will be a function of the initial deflection $u_{z0} = u_{z0}(x, 0)$.

Let us consider the first approximation:

$$f(x, \ t) = f(x) = -Mx, \ \ 0 \leq x \leq l, \tag{19}$$

where $M$ is some positive constant.

The negative sign corresponds to a decrease in the heating of the plate intensity from the sealing to the free edge (with an increase in the $x$ coordinate) due to the initial deflection. Then, the expression (18) can be converted to the following form:

$$T(x, \ z, \ t) = Cz\frac{t}{t + a} - Mx + T_0, \ \ 0 \leq x \leq l, \ \ 0 \leq z \leq h, \ \ t > 0. \tag{20}$$

At $C$ = 200 K/m, $a$ = 1 s, $M$ = 3 K/m and the initial deflection $u_{z0}$ = –0,1 $x^2$ the temperature field of the plate is shown in Figure 4.

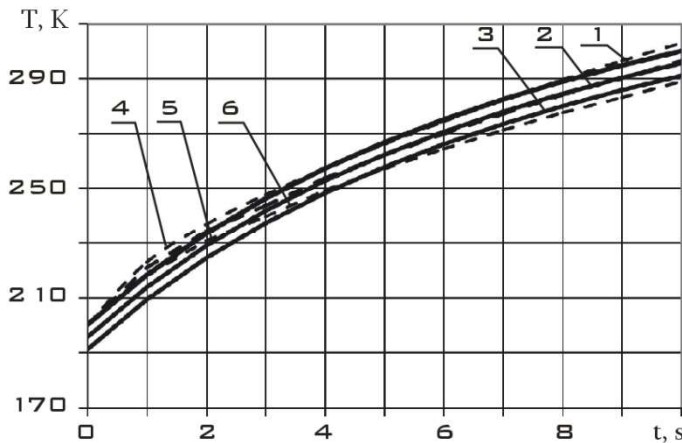

**Figure 4.** Temperature dynamics of the plate surface layer at the initial deflection $u_{z0}$ = –0,1 $x^2$, received as a result of the numerical simulation (dotted lines): 1 close to the seal ($x \approx 0$); 2 in the middle of the plate ($x \approx 1/2$); 3 on the free edge of the plate ($x \approx 1$); according to the formula (3.18) at $C$ = 200 K/m, a = 1 s and M = 3 K/м (solid lines); 4 close to the seal ($x \approx 0$); 5 in the middle of the plate ($x \approx 1/2$); 6 on the free edge of the plate ($x \approx 1$).

From the analysis of Figure 4, it can be argued about the good convergence of the computational experiment results (ANSYS package) and the approximate analytical expression (20) for the temperature field. This means that Equation (2) is completely satisfied by the approximate expressions (16) and (20). Let us further consider the second component of the displacement vector, $u_y = u_y(x, y, t)$. Instead of (9), let us take the following expression for $u_y$:

$$u_y(x, \ y, \ t) = u_{1y}(x)y + u_{2y}(\ t).\tag{21}$$

Then, the second mixed derivative with respect to the expression (21) will have the following form:

$$\frac{\partial^2 u_y(x, \ y, \ t)}{\partial x\, \partial y} = \frac{du_{1y}(x)}{dx}.\tag{22}$$

In accordance with the first equation of the system (10), we can write the following:

$$\frac{\partial^2 u_y(x, \ y, \ t)}{\partial x\, \partial y} = \alpha\frac{\partial T(x, \ z, \ t)}{\partial x}.\tag{23}$$

Taking into account the expression for temperature (20) and derivative (22), Equation (23) can be rewritten as follows:

$$\frac{du_{1y}(x)}{dx} = -\alpha M.\tag{24}$$

Or after integration (22):

$$u_{1y}(x) = -\alpha Mx + C_1,\tag{25}$$

where $C_1$ is some constant.

Then, expression (21) will take the following form:

$$u_y(x, \ y, \ t) = (-\alpha Mx + C_1)y + u_{2y}(\ t).\tag{26}$$

Let us return to Equation (11). For the expression (26), the derivatives included in Equation (11) can be written as follows:

$$\frac{\partial^3 u_y(x, y, t)}{\partial x^3} = 0; \quad \frac{\partial^2 u_y(x, y, t)}{\partial t^2} = \frac{d^2 u_{2y}(t)}{dt^2}. \tag{27}$$

Taking into account expressions (20) and (27), Equation (11) will take the following form:

$$\frac{d^2 u_{2y}(t)}{dt^2} = \frac{2}{hb}\frac{E}{\rho}\alpha \int_0^h \left[Cz\frac{t}{t+a} - Mx\right]dz, \quad 0 \le x \le l, \ t > 0. \tag{28}$$

After taking the integral in the right part (28), we will have the following:

$$\frac{d^2 u_{2y}(t)}{dt^2} = \frac{1}{b}\frac{E}{\rho}\alpha\left(Ch\frac{t}{t+a} - 2Mx\right), \quad 0 \le x \le l, \ t > 0. \tag{29}$$

Let us simplify (29) by neglecting the second term in brackets on the right side. The results of numerical modeling (Figure 4) allow such a simplification.

Let us integrate (29) twice with respect to $t$ and, taking into account the simplification, we will obtain the following:

$$u_{2y}(t) = \alpha\frac{Ch}{b}\frac{E}{\rho}\left\{\frac{t^2}{2} - a(t+a)[\ln(t+a) - 1]\right\} + C_2 t + C_3, \quad t > 0, \tag{30}$$

where $C_2$ and $C_3$ are some integration constants.

Then, the expression (26) will take the following form:

$$u_y(x,y,t) = \alpha\left[(-Mx + C_1)y + \frac{Ch}{b}\frac{E}{\rho}\left\{\frac{t^2}{2} - a(t+a)[\ln(t+a) - 1]\right\}\right] + C_2 t + C_3,$$
$$x \approx 0, \ 0 \le y \le \frac{b}{2}, t > 0. \tag{31}$$

In (31), the range $x \approx 0$ means that the solution is correct near the seal at small values of the x coordinate. These values should allow a negligible smallness of the second term in brackets on the right side of the equation (29) compared to the first term.

Let us investigate if the expression (31) satisfies the boundary conditions (12), (14), and the initial condition (15). Let us use parameter values $C = 200$ K/m, $a = 1$ s, $M = 3$ K/m. They were obtained in [4] when solving the one-dimensional problem and adopted when constructing the temperature field (Figure 4). Then, the expression (31) is converted to the following form:

$$u_y(x,y, 0) = \alpha(-Mx + C_1)y + C_3 = 0, \quad x \approx 0, \ 0 \le y \le \frac{b}{2}, t = 0. \tag{32}$$

In this case, for $x \approx 0$, it is necessary to put $C_1 = C_3 = 0$ to satisfy the initial condition (2.21). Then, we can specify the type of solution (31):

$$u_y(x,y,t) = \alpha\left[-Mxy + \frac{Ch}{b}\frac{E}{\rho}\left\{\frac{t^2}{2} - a(t+a)[\ln(t+a) - 1]\right\}\right] + C_2 t,$$
$$x \approx 0, \ 0 \le y \le \frac{b}{2}, t > 0. \tag{33}$$

It is incorrect to demand satisfaction of the static boundary conditions (14) for $x \approx 0$. Therefore, let us limit ourselves to checking the geometric boundary conditions (12). Taking into account the expression (33), they will have the following form:

$$\begin{cases} \alpha\frac{Ch}{b}\frac{E}{\rho}\left\{\frac{t^2}{2} - a(t+a)[\ln(t+a) - 1]\right\} + C_2 t = 0, & t > 0 \\ -\alpha My = 0, & 0 \le y \le \frac{b}{2} \end{cases}. \tag{34}$$

These conditions can only be met at $t \approx 0$ and $y \approx 0$.

Thus, when solving the one-dimensional problem of thermal conductivity [4], a solution that fully satisfies all the boundary and initial conditions was obtained. However, Equation (11) was valid only for $t \approx 0$ [4]. The approximate solution of the two-dimensional

thermoelasticity problem (33) satisfies the initial condition (15) at $x \approx 0$ and the boundary condition (12) at $t \approx 0$ and $y \approx 0$.

However, it makes Equation (11) valid for $x \approx 0$ for arbitrary values of other coordinates.

## 3. Simulation Part

To assess the adequacy of the simulation results, a computational experiment was conducted in the ANSYS package for a plate, the main parameters of which are given in Table 1 [4,18].

**Table 1.** The main parameters of the simulated plate.

| Parameter | Designation | Value | Dimension |
|---|---|---|---|
| Solar panel frame material | – | MA2 | – |
| Coefficient of thermal conductivity | $\lambda$ | 96.3 | W/(m·K) |
| Stefan–Boltzmann constant | $\Theta$ | $5.67 \times 10^{-8}$ | W/(m²·K⁴) |
| External heat flux | $Q$ | 1400 | W/m² |
| Vacuum temperature | $T_c$ | 3 | K |
| Initial temperature of the solar panel frame | $T_0 = T(z, 0)$ | 200 | K |
| Degree of blackness | $e$ | 0.2 | – |
| Specific heat | $c$ | 1130.4 | J/(kg·K) |
| Density | $\rho$ | 1780 | kg/m³ |
| Young's Module | $E$ | $4 \times 10^{10}$ | Pa |
| Shift modulus | $\mu$ | $1.6 \times 10^{10}$ | Pa |
| Poisson's Ratio | $\nu$ | 0.3 | – |
| Solar panel length | $l$ | 1 | m |
| Solar panel width | $b$ | 0.5 | m |
| Solar panel frame thickness | $h$ | 1 | mm |
| Equation of the initial deflection of the plate | $u_{z0}$ | $-0.1\,x^2$ | m |

The general view of the plate at the moment of temperature shock in the ANSYS finite element model is shown in Figure 5 [18].

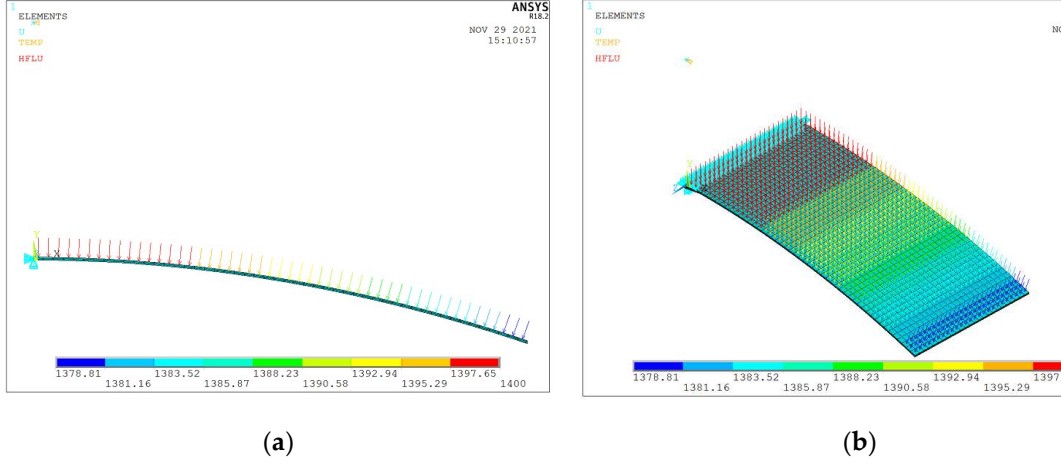

      (**a**)            (**b**)

**Figure 5.** The general view of the plate with initial deflection at the moment of temperature shock [18]. (**a**) profile view; (**b**) axonometric view.

A comparison of the computational experiment results and the approximate dependence for deflections (16) is shown in Figure 6.

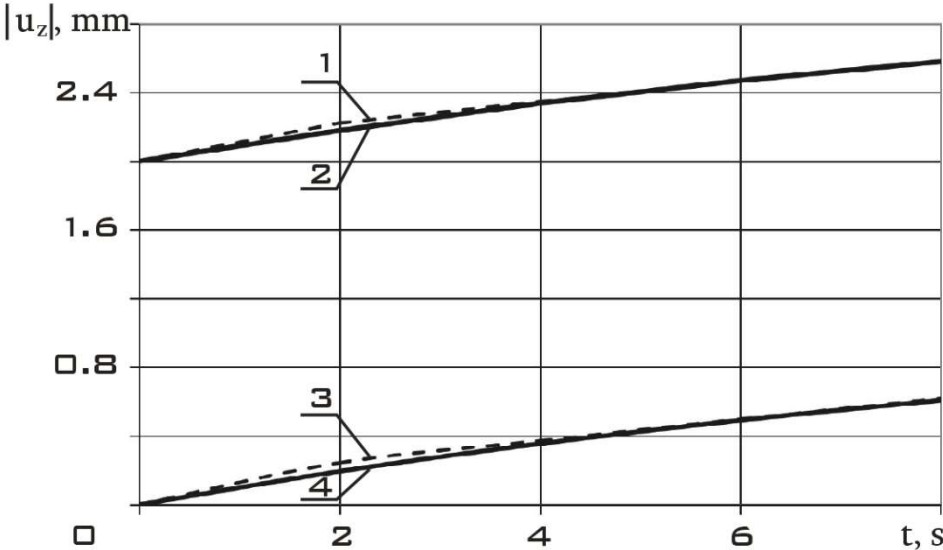

**Figure 6.** The dynamics of deflection modules $u_z$ of the plate end-section ($x = l$) without initial deflection: 1 based on the results of the computational experiment; 2 by the approximate dependence (16); at the initial deflection $u_{z0} = -0.1\,x^2$; 3 based on the results of the computational experiment; 4 by the approximate dependence (16).

The analysis of Figure 6 shows good convergence of the results.

## 4. Conclusions

Thus, this paper presents an approximate numerical solution of the static two-dimensional thermoelasticity problem for the temperature shock of a thin plate. The approximate dependencies of the components of the displacement vector are obtained. An approximate analytical representation of the temperature field of the plate is presented and justified. The dependencies obtained in the work are compared with the results of the computational experiment in the ANSYS package. This comparison showed good convergence of the results. The materials from the work can be used both for practical purposes and for theoretical analysis; for example, when assessing the effect of the temperature shock on the motion of a small spacecraft. The proposed approach is of great practical importance. It makes it possible to effectively assess the temperature deformations of large elastic elements of the spacecraft, investigate their significance, and develop control laws to reduce the influence of the temperature shock. The theoretical development is connected with the complication of the model by abandoning a number of restrictions used in the static two-dimensional formulation of the thermoelasticity problem.

**Author Contributions:** Conceptualization, A.S. and V.S.; methodology, A.S. and D.O.; software, V.S. and A.N.; validation, A.S., V.S. and D.O.; formal analysis, A.S.; investigation, A.S. and D.O.; resources, A.S. and V.S.; data curation, A.S. and A.N.; writing—original draft preparation, A.S.; writing—review and editing, A.S., V.S. and D.O.; visualization, V.S.; supervision, A.S.; project administration, A.S.; funding acquisition, A.S., D.O. and A.N. All authors have read and agreed to the published version of the manuscript.

**Funding:** This work is supported by the Ministry of Education and Science of the Russian Federation in the framework of the State Assignments to higher education institutions and research organizations in the field of scientific activity (the project FSSS-2023-0007).

**Data Availability Statement:** Not applicable.

**Conflicts of Interest:** The authors declare no conflict of interest.

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
