# Peer review of "Investigating the Temperature Shock of a Plate in the Framework of a Static Two-Dimensional Formulation of the Thermoelasticity Problem"

_aerospace, doi:10.3390/aerospace10050445_

Round 1
Reviewer 1 Report
The manuscript "Investigating the temperature shock of a plate in the framework of a static two-dimensional formulation of the thermoelasticity problem" could be interesting from a theoretical point of view. However, the authors did not avoid serious flaws, which do not allow me to accept it as a scientific article.
First and foremost: the manuscript is based on a heat transfer model, which is badly stated. Figure one shows heat flux, but the mathematical model does not consider heat flux in its description. It is a crucial fault because the heat flux depends on the temperature gradient and heat transfer coefficient, which is not considered at all.
Furthermore, the "computational experiment" (the term which confuses me) is not described at all. A scientific paper should describe the methods and models in a way that makes it possible to conduct the same study by another researcher and obtain the same results. The description presented in the paper does not allow it.
Regretfully, considering the remarks mentioned above, I do not recommend the manuscript for publication.
Author Response
First and foremost: the manuscript is based on a heat transfer model, which is badly stated. Figure one shows heat flux, but the mathematical model does not consider heat flux in its description. It is a crucial fault because the heat flux depends on the temperature gradient and heat transfer coefficient, which is not considered at all.
This is not true at all. Heat flux is not internal. It is an external heat flux. It is taken into account in the boundary conditions of the problem (the second equation of the system (1) is the first term of the right-hand side). The first figure shows the heat flux outside the body, not inside it. This is important for describing the heat conduction model. This flux is not in the heat conduction equation because it is not internal. There is no error here.
Furthermore, the "computational experiment" (the term which confuses me) is not described at all. A scientific paper should describe the methods and models in a way that makes it possible to conduct the same study by another researcher and obtain the same results. The description presented in the paper does not allow it.
The authors agree with this remark. Thank you. The details of the experiment have been added to the work.
Reviewer 2 Report
This manuscript describes the stress-strain state of a homogeneous rectangular plate after a temperature shock. To study the stress-strain state of the plate, a two-dimensional thermoelasticity problem is posed for static formulation. Some issues in the manuscript are expected to be solved. I suggest the following changes and improvements:
-In the abstract, the authors should consider adding more qualitative and quantitative results and important findings. In addition, the innovation of the evaluation method proposed in this paper should be further highlighted.
-The current structure of the introduction needs to be better organized. Additionally, the authors need to be improved the last part of the introduction considering the main theme/objectives and findings of the study.
- The introduction part needs to be extended by discussing more relevant papers. The authors should appropriately extend this section by discussing more relevant works focusing on different methods and models in the literature. For example, it is suggested to read and discuss the following relevant works:
Reliability based topology optimization of thermoelastic structures using bi-directional evolutionary structural optimization method. Int J Mech Mater Des (2023). https://doi.org/10.1007/s10999-023-09641-0
Displacement minimization of thermoelastic structures by evolutionary thickness design. Comput. Methods Appl. Mech. Eng. 179, 361–378 (1999). https://doi.org/10.1016/S0045-7825(99)00047-X
- It will be more appropriate to add a paragraph at the end of the introduction section illustrating the paper's layout.
- Please declare the assumptions and limitations of the model. Large or small displacement, geometrically linear or nonlinear….
-Please elaborate on the ANSYS modeling details. It needs to be clearly shown in the manuscript how the temperature was considered in the simulation. Element type and mesh size study need to be included.
In conclusion, please clearly mention the practical outcome of the study. Please clarify what the employed technique offers to the engineering field compared to other methods in the literature.
Author Response
In the abstract, the authors should consider adding more qualitative and quantitative results and important findings. In addition, the innovation of the evaluation method proposed in this paper should be further highlighted.
Thank you. Added to the abstract.
The current structure of the introduction needs to be better organized. Additionally, the authors need to be improved the last part of the introduction considering the main theme/objectives and findings of the study.
Thank you. Improved.
The introduction part needs to be extended by discussing more relevant papers. The authors should appropriately extend this section by discussing more relevant works focusing on different methods and models in the literature.
Thank you. Expanded.
For example, it is suggested to read and discuss the following relevant works:
Reliability based topology optimization of thermoelastic structures using bi-directional evolutionary structural optimization method. Int J Mech Mater Des (2023). https://doi.org/10.1007/s10999-023-09641-0
Displacement minimization of thermoelastic structures by evolutionary thickness design. Comput. Methods Appl. Mech. Eng. 179, 361–378 (1999). https://doi.org/10.1016/S0045-7825(99)00047-X
Discussion of work added.
It will be more appropriate to add a paragraph at the end of the introduction section illustrating the paper's layout.
The paragraph has been added.
Please declare the assumptions and limitations of the model. Large or small displacement, geometrically linear or nonlinear….
Assumption 12 has been added separately for the initial deflection. It does not refer to any specific meaning. However, the buckling of the plate during thermal shock must not be allowed.
Please elaborate on the ANSYS modeling details. It needs to be clearly shown in the manuscript how the temperature was considered in the simulation. Element type and mesh size study need to be included.
Thank you. Added a figure with a finite element model. Added table with input data for modeling.
In conclusion, please clearly mention the practical outcome of the study. Please clarify what the employed technique offers to the engineering field compared to other methods in the literature.
Thank you. Added.
Reviewer 3 Report
Stress-strain state of a homogeneous rectangular plate after a temperature shock have been discussed via an approximate solution. Approximate dependences of the components of the displacement vector are obtained. An approximate analytical representation of the temperature field of the plate has been obtained and presented. Some parametric results have been supplied.
Manuscript can be accepted for publication after minor revision:
1. Which assumptions have been taken into consideration for heat conduction equation?
2.The novelty of the present paper then authors other /listed below)paper can be mentioned in clearly.
Investigation of the Stress-Strain State of a Rectangular Plate after a Temperature Shock
3. Some related references about the thermal effect and plates must be cited:
Two-Scale Asymptotic Homogenization Method for Composite Kirchhoff Plates with in-Plane Periodicity.Aerospace 2022, 9(12), 751; https://doi.org/10.3390/aerospace9120751
Abouelregal, A.E., Akgöz, B., Civalek, O., Magneto-thermoelastic interactions in an unbounded orthotropic viscoelastic solid under the Hall current effect by the fourth-order Moore-Gibson-Thompson equation. Computers and Mathematics with Applications, 141 (2023) 102-115.
4-There exist some equations which neither are referred to a scientific reference nor are developed in the present study. They certainly require appropriate references.
Author Response
- Which assumptions have been taken into consideration for heat conduction equation?
The assumptions are listed separately in the assumptions section. Assumption 12 for initial deflection has been added. It does not refer to any particular value. However, it should not allow for loss of plate stability during temperature shock.
2. The novelty of the present paper then authors other /listed below)paper can be mentioned in clearly.
Thank you. Novelty is highlighted in the introduction.
3. Some related references about the thermal effect and plates must be cited:
Thank you. References added.
There exist some equations which neither are referred to a scientific reference nor are developed in the present study. They certainly require appropriate references.
Thank you. References added.
Round 2
Reviewer 1 Report
The authors dealt with all the issues of the manuscript.
Reviewer 2 Report
I suggest accepting the manuscript.
Reviewer 3 Report
accepted